# Multi-Stage Manipulation with Demonstration-Augmented Reward, Policy, and World Model Learning

**Adrià López Escoriza** [1 2]   **Nicklas Hansen** [1]   **Stone Tao** [1 3]   **Tongzhou Mu** [1]   **Hao Su** [1 3]

## Abstract

Long-horizon tasks in robotic manipulation present significant challenges in reinforcement learning (RL) due to the difficulty of designing dense reward functions and effectively exploring the expansive state-action space. However, despite a lack of dense rewards, these tasks often have a multi-stage structure, which can be leveraged to decompose the overall objective into manageable subgoals. In this work, we propose Demonstration-Augmented Reward, Policy, and World Model Learning (DEMO³), a framework that exploits this structure for efficient learning from visual inputs. Specifically, our approach incorporates multi-stage dense reward learning, a bi-phasic training scheme, and world model learning into a carefully designed demonstration-augmented RL framework that strongly mitigates the challenge of exploration in long-horizon tasks. Our evaluations demonstrate that our method improves data-efficiency by an average of $40\%$ and by $70\%$ on particularly difficult tasks compared to state-of-the-art approaches. We validate this across 16 sparse-reward tasks spanning four domains, including challenging humanoid visual control tasks using as few as five demonstrations. Website with code and visualizations can be found at **https://adrialopezescoriza.github.io/demo3**.

## 1. Introduction

Reinforcement learning (RL) with dense rewards has enabled significant progress in high-dimensional control tasks. Many such tasks are now solvable when the reward function is carefully designed for the specific goal. In particular, model-based RL (MBRL) has demonstrated strong performance in these high-dimensional problems (Ha &

Schmidhuber, 2018; Zhang et al., 2018; Kidambi et al., 2020; Hafner et al., 2020; Yu et al., 2020; Hansen et al., 2022; 2024; Sferrazza et al., 2024). However, designing accurate reward functions is challenging. Poorly designed rewards can lead agents to become trapped in local minima or exploit unintended shortcuts, resulting in undesirable behaviors (Clark & Amodei, 2016). More critically, scaling reward design to complex tasks is highly impractical: the larger the state space and the longer the horizon, the more intricate the reward must be. While recent approaches leveraging Large Language Models (Kwon et al., 2023; Ma et al., 2023; Xie et al., 2024) and Vision-Language Models (Rocamonde et al., 2024; Baumli et al., 2024) for reward generation show promise, they still struggle with high-precision requirements, particularly in manipulation problems. In contrast, sparse rewards, such as binary signals indicating task or subtask completion, are much easier to obtain. However, traditional RL methods still struggle to learn effectively from sparse rewards.

Fortunately, long-horizon tasks do offer opportunities to simplify the problem. Typically, such tasks exhibit a natural multi-stage structure. For example, a pick-and-place task can be broken down into subtasks such as grasping, lifting, and placing. Each of these can be associated with stage indicators or rewards that can easily be queried from the environment. This multi-stage structure allows these tasks to be decomposed into more manageable subgoals, enabling the agent to collect rewards more frequently (Smith et al., 2020; Di Palo & Johns, 2021). However, subgoal sparse rewards can still be insufficient if the distance between subgoals is too great, leading back to the exploration problem.

Prior work shows that Learning from Demonstrations (LfD) can help mitigate exploration issues in sparse reward settings. Algorithms such as CoDER (Zhan et al., 2022) and MoDem (Hansen et al., 2023; Lancaster et al., 2024) leverage demonstrations to populate the replay buffer of an off-policy RL algorithm (Sutton & Barto, 2018), but they often scale poorly with task complexity as they need demonstrations that sufficiently cover the behavior space. Inverse RL methods (Trott et al., 2019; Wu et al., 2021; Memarian et al., 2021; Escontrela et al., 2022) train RL on a reward function that is learned from demonstrations, but inverse RL alone

---

[1]University of California San Diego [2]ETH Zürich, Switzerland [3]Hillbot. Correspondence to: Adrià López Escoriza <alopez@ethz.ch>.

*Proceedings of the 42^{nd} International Conference on Machine Learning*, Vancouver, Canada. PMLR 267, 2025. Copyright 2025 by the author(s).

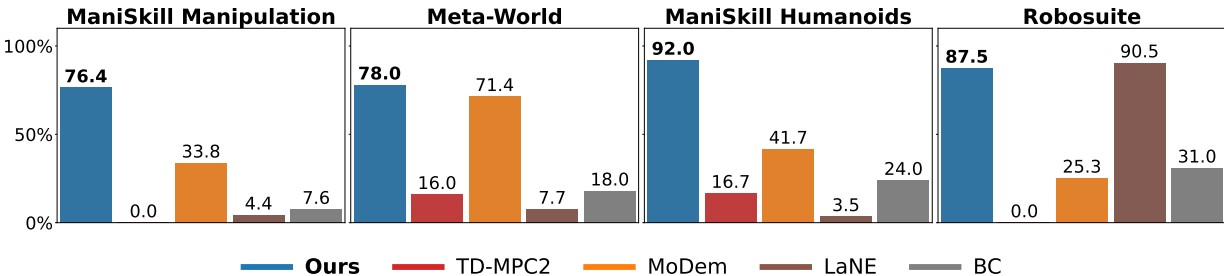

Figure 1. **Summary of results**. Final success rate (%) achieved by our method and a set of strong baselines, averaged across all tasks within each of 4 domains. Average of 5 seeds. Given a handful of demonstrations, our method achieves high success rates in challenging visual manipulation tasks with sparse rewards, far exceeding previous state-of-the-art methods. See Appendix A for per-task results.

often struggles as the learned reward function can have poor predictions on unseen states. Lastly, these methods typically require a vast amount of samples to learn a reward function (Kumar et al., 2022; Mu et al., 2024) before any policy learning can begin.

In this paper, we build on demonstration-augmented RL to tackle multi-stage manipulation tasks with sparse stage-wise rewards. We introduce DEMO³, a model-based RL algorithm that leverages a limited number of demonstrations for three key purposes: **learning a policy, a world model, and a dense reward**. DEMO³ exploits the multi-stage structure of long-horizon tasks to transform sparse stage indicators into a stage-wise dense reward. This enables dense feedback in a structured way, prioritizing achieving subgoals over following demonstrations. Unlike prior work, our dense reward is learned online, alongside policy and world model learning.

We evaluate our method on a range of challenging manipulation tasks from Meta-World (Yu et al., 2021), Robosuite (Zhu et al., 2022), as well as both humanoid and tabletop manipulation tasks from ManiSkill3 (Tao et al., 2024b). Our results (see Figure 1) demonstrate that our method outperforms state-of-the-art methods by an average of 40%, and for more complex, longer-horizon tasks, this performance gap increases to 70%. **Our main contributions can be summarized as follows:**

1. We introduce DEMO³, an MBRL algorithm for highly data-efficient robotic manipulation from visual inputs and sparse rewards. Our method integrates online dense reward learning into RL for multi-stage tasks.

2. We conduct extensive experiments in 16 tasks across 4 domains to demonstrate the data-efficiency and robustness of our approach compared to existing methods.

3. We analyze the relative importance of each component of our framework, and are **open-sourcing all code and demonstrations used in this work**.

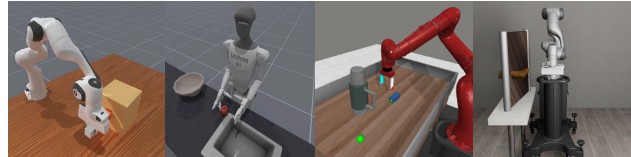

Figure 2. **Task domains**. We evaluate methods on **16** multi-stage image-based sparse-reward tasks spanning four domains: Meta-World (Yu et al., 2021), Robosuite (Zhu et al., 2022), as well as manipulation and humanoid tasks from ManiSkill3 (Tao et al., 2024b). See Appendix D for a complete overview of tasks.

## 2. Preliminaries

**Problem formulation**. We aim to learn control policies for multi-stage, long-horizon tasks, which we model as infinite-horizon Markov Decision Processes (Bellman, 1957) defined by the tuple $(\mathcal{S}, \mathcal{A}, \mathcal{P}, \mathcal{R}, \gamma)$. Here, $\mathcal{S}$ and $\mathcal{A}$ denote the state and action spaces, respectively, $\mathcal{P}$ is the (unknown) state transition probability function, $\mathcal{R}$ is a sparse reward function, and $\gamma \in [0, 1)$ is the discount factor. Our goal is to learn a policy $\pi_\theta : \mathcal{S} \to \mathcal{A}$ parameterized by $\theta$ that maximizes the expected cumulative reward (return) over an infinite time horizon, formalized as $\max \mathbb{E}_{\pi_\theta} \left[ \sum_{t=0}^{\infty} \gamma^t r_t \right]$.

**Multi-stage sparse rewards**. In this work, we focus on *multi-stage* tasks where the overall objective can be decomposed into a sequence of $N$ subgoals or stages. Stage indicators are often easy to obtain; for example, in manipulation tasks, it is straightforward to query whether the agent has grasped an object. We model the (sparse) reward as a stage indicator function $r : \mathcal{S} \to \{1, 2, \ldots, N\}$ that maps each state to its corresponding stage. We assume no access to any privileged information from the environment (e.g. object configurations), and instead consider *multi-modal observations* $\mathbf{o} = (\mathbf{x}, \mathbf{q})$ where $\mathbf{x}$ denotes raw RGB images coming from the agent's cameras, and $\mathbf{q}$ denotes the proprioceptive state of the robot. This constitutes realistic sensory inputs available in typical robotic platforms.

**TD-MPC2** (Hansen et al., 2022; 2024) is a model-based RL algorithm that combines Model Predictive Control (MPC), a learned latent-space world model, and a terminal value function learned via temporal difference (TD) learning. Specif-

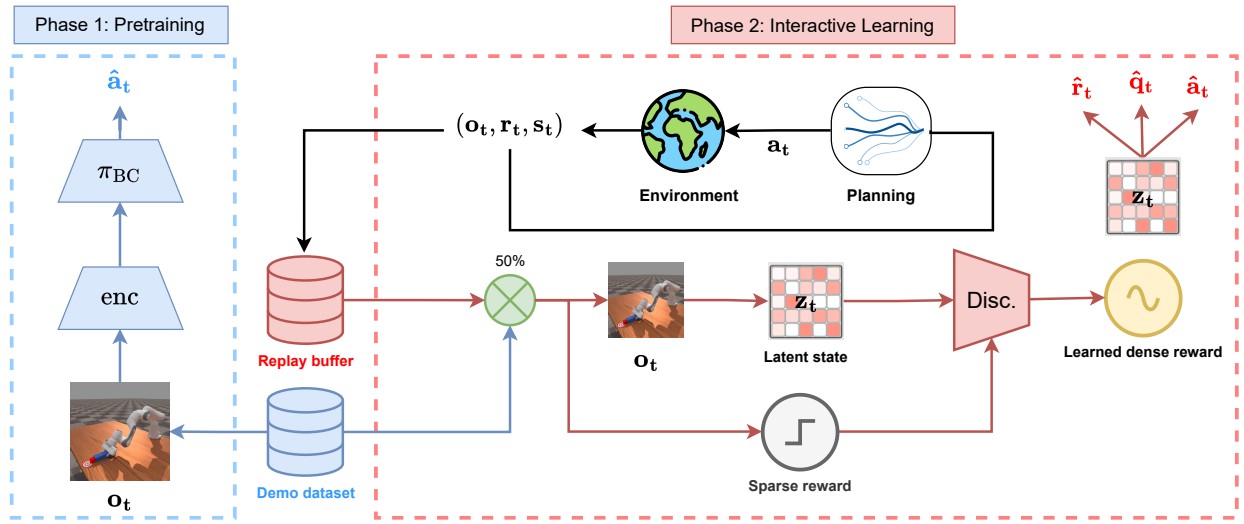

*Figure 3.* **Method overview**. We present a two-phase framework for multi-stage visual manipulation from sparse rewards that leverages a handful of demonstrations for dense reward learning and MBRL. **Phase 1 (*left*):** policy and encoder is pre-trained on the available demonstrations using behavioral cloning, which serve as initialization for the next phase. **Phase 2 (*right*):** the agent iteratively collects environment data via planning and uses all available data to update its world model as well as a latent state discriminator; this discriminator is used to transform sparse environment rewards into a learned dense reward for world model learning and subsequent planning.

ically, TD-MPC2 learns a representation $\mathbf{z} = h_\theta(\mathbf{o})$ that maps a high-dimensional observation $\mathbf{o}$ into a compact representation $\mathbf{z}$, as well as a dynamics model in this latent space $\mathbf{z}' = d_\theta(\mathbf{z}, \mathbf{a})$. In addition, TD-MPC2 also learns prediction heads, $R_\theta, Q_\theta, \pi_\theta$, for *(i)* instantaneous reward $r = R_\theta(\mathbf{z}, \mathbf{a})$, *(ii)* state-action value $Q_\theta(\mathbf{z}, \mathbf{a})$, and *(iii)* a policy prior $\mathbf{a} \sim \pi_\theta(z)$. The policy prior $\pi_\theta$ serves to "guide" planning towards high-return trajectories and is optimized to maximize temporally weighted $Q$-values. The remaining components are jointly optimized to minimize TD-errors, and reward and latent state prediction errors, minimizing

$$\mathcal{L}_{\text{TD-MPC}}(\theta) = \sum_{i=t}^{t+H} \lambda^{i-t} \left[ \mathcal{L}_Q(\theta) + \mathcal{L}_R(\theta) + \mathcal{L}_h(\theta) \right], \quad (1)$$

where $\mathbf{o}'_t, \mathbf{z}'_t$ are the (latent) states at time $t+1$. During environment interaction, TD-MPC2 selects actions via sample-based planner MPPI (Williams et al., 2015) and the learned world model. We adopt TD-MPC2 as our choice of visual MBRL algorithm due to its simplicity and strong empirical performance but emphasize that our framework can be instantiated with any MBRL algorithm.

## 3. Method

In this work, we address the challenge of solving multi-stage manipulation tasks from sparse rewards. Such long-horizon tasks are particularly difficult due to the combinatorial complexity of the state-action space and the lack of informative feedback across extended horizons. To overcome these issues, we propose DEMO[3], a novel RL method that uses demonstrations for a three-fold purpose: to learn a policy, a world model, and a dense reward function simultaneously.

As our main algorithmic contribution, we introduce stage-specific reward learning. In particular, we extend the strategy on reward learning from demonstrations (LfD) presented in Mu et al. (2024) to online reward learning within a world model. By learning structured, multi-stage rewards online alongside world model and policy, our method provides more frequent and meaningful training signals to the agent than prior work on demonstration-augmented RL.

Our approach builds directly upon the strengths of prior work. In particular, we leverage MoDem's multi-phase accelerated learning framework and use TD-MPC2 as our backbone for its robustness and generalizability.

### 3.1. Model-based RL with online reward learning

Sparse rewards are a major challenge in RL, particularly for long-horizon tasks comprising multiple stages. To overcome this, we learn to densify sparse rewards with a small number of demonstrations. For this, we introduce a series of discriminators $\{\delta_k\}_{k=0}^N$, each corresponding to a task stage $k \in \{0 \dots N\}$. The objective of each discriminator is to predict the likelihood of progressing to the next stage based on the latent state representation $\mathbf{z_t}$ produced by the back-bone world model.

Therefore, each discriminator $\delta_k$ acts as a **stage classifier** trained to distinguish states as either leading or not leading to successful stage transitions. For each stage $k$, we use a typical Binary Cross Entropy (BCE) loss:

$$\mathcal{L}_{\delta_k} = \mathop{\mathbb{E}}_{(\mathbf{o_t}, r_t = k, s_t) \sim \mathcal{B}} \left[ \text{BCE}(\mathbb{1}_{s_t > k}, \delta_k(\mathbf{h}(\mathbf{o_t}))) \right], \quad (2)$$

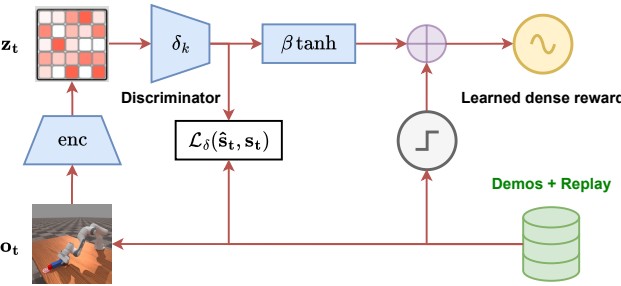

*Figure 4.* **Dense reward learning**. At each update step, the continuous output of a stage discriminator is added to the environment sparse reward. The discriminator output is normalized to the $[-\beta, \beta]$ interval with a $\tanh$ operator.

where h denotes the world model encoder, $\mathcal{B}$ is the replay buffer, and $s_t$ represents the maximum stage that will be reached by the trajectory after the given sample:

$$s_t = \max_{t' \geq t} r_{t'}, \tag{3}$$

We refer to $s_t$ as the *maximum stage label* of a sub-trajectory. Thus, each trajectory, $\boldsymbol{\tau_i} = \{(\mathbf{o_t}, \mathbf{a_t}, r_t, \mathbf{o_{t+1}}, s_t)\}_{t=0}^{T-1}$, is annotated with *maximum stage labels* $s_t$ that serve as success labels for the stage discriminators. Then, as presented in Algorithm 1, at each update of the world model, the discriminators are updated as an additional part of the model. Specifically, for a given sample from the replay buffer $(\mathbf{o_t}, \mathbf{a_t}, \mathbf{o_{t+1}}, r_t, s_t) \sim \mathcal{B}$, the sparse reward associated with a given stage, $k = r_t$, will tell us which discriminator, $\delta_k$, will be updated by that sample. If the *maximum stage label* $s_t$ is greater than the current stage reward $k = r_T$, the sample belonging to a trajectory with a successful stage transition will be treated as a positive example in the classifier loss. Note that in the event that no samples with a stage reward, $r_t = k$, would appear in a given batch, the discriminator $\delta_k$ for that stage would simply not get updated at that step. Therefore, while the algorithm is capable of working without any demonstrations, using a small demonstration dataset can significantly accelerate the training of the world model and discriminators.

While training the world model, the discriminators are used to **generate dense rewards** as per

$$\hat{r}_t^\delta = r_t + \beta \cdot \tanh(\delta_{r_t}(\mathbf{z_t})) \tag{4}$$

where the output of the discriminator is mapped to the $[-\beta, \beta]$ interval. The process is illustrated by Figure 4. We set $\beta$ to be a hyperparameter with $\beta \leq 1/3$ to ensure that rewards never cross between different stage regions. This is to ensure that states belonging to a more advanced stage always get a higher reward than lower-stage states. Effectively, our method rewards states that have a higher chance of transitioning to the next stage and penalizes those

---

**Algorithm 1** DEMO³ (**Phase 2**)

**Require:** Demonstration dataset $\mathcal{D}$, number of stages $N$
1: Initialize discriminators $\{\delta_k\}_{k=0}^N$
2: Initialize replay buffer $\mathcal{B} \leftarrow \{\varnothing\}$
   **Rollout**
3: **for** each environment step **do**
4:     **if** rand() $\geq \alpha$ **then**
5:         Agent step: $\mathbf{a_t} \sim \pi_{\mathrm{BC}}(\mathbf{a}|\mathrm{h}(\mathbf{o_t}))$
6:     **else**
7:         Agent step: $\mathbf{a_t} \leftarrow \mathrm{WM}_{\mathrm{plan}}(\mathbf{o_t})$
8:     **end if**
9:     Env step: $(\mathbf{o_{t+1}}, r_t) \leftarrow \mathrm{Env}(\mathbf{o_t}, \mathbf{a_t})$
10:    Save sample: $\tau \leftarrow \tau \cup (\mathbf{o_t}, \mathbf{a_t}, \mathbf{o_{t+1}}, r_t)$
11:    **if** episode done **then**
12:        Reset environment
13:        Compute maximum stage labels: $\{s_t\}_0^T$
14:        Save trajectory: $\mathcal{B} \leftarrow \mathcal{B} \cup (\tau \cup \{s_t\}_0^T)$
15:    **end if**
16:    $\alpha \leftarrow \min(1, \alpha_0 \cdot t)$
17: **end for**
   **Update**
18: **for** each update step **do**
19:    Sample: $\left\{(\mathbf{o_t}, \mathbf{o_{t+1}}, \mathbf{a_t}, r_t, s_t)_{t_0}^{t_0+H}\right\} \sim (\mathcal{B} \cup \mathcal{D})$
20:    Predict dense reward $(\hat{r}_t^\delta)_{t_0}^{t+H}$
21:    Compute world model losses: $\mathcal{L}_R, \mathcal{L}_Q, \mathcal{L}_h, \mathcal{L}_\pi$
22:    Compute discriminator loss: $\mathcal{L}_\delta = \frac{1}{N} \sum_k \mathcal{L}_\delta^k$
23:    Gradient step: $\theta \leftarrow \theta + \rho \nabla \mathcal{L}_P$
24: **end for**

---

that do not, encouraging the agent to explore regions with a higher probability of transition.

The total world model loss integrating these signals becomes

$$\mathcal{L}_P = \mathcal{L}_R + \mathcal{L}_Q + \mathcal{L}_h + \mathcal{L}_\delta, \tag{5}$$

where $\mathcal{L}_R$, $\mathcal{L}_Q$, and $\mathcal{L}_h$ represent the TD-MPC2 world model losses: *(i)* reconstruction, *(ii)* value estimation, and *(iii)* latent dynamics losses. Importantly, $\mathcal{L}_R$ is computed to predict the learned dense reward produced by the discriminator, $\hat{r}_t^\delta$, thus providing a richer learning signal than pure sparse rewards. Finally, $\mathcal{L}_\delta$ is the average loss of all the stage discriminators. As in Hansen et al. (2024), the total loss is used to compute gradients for the world model, meaning that $\mathcal{L}_\delta$ is also used to learn the observation encoder.

### 3.2. Training scheme

In order to further boost the data-efficiency of DEMO³, we build upon previous work on accelerating MBRL with demonstrations. Specifically, we draw inspiration from Mo-Dem (Hansen et al., 2023) and propose a bi-phase training scheme in which we first use demonstrations to pre-train an initial policy through behavioral cloning (Atkeson & Schaal,

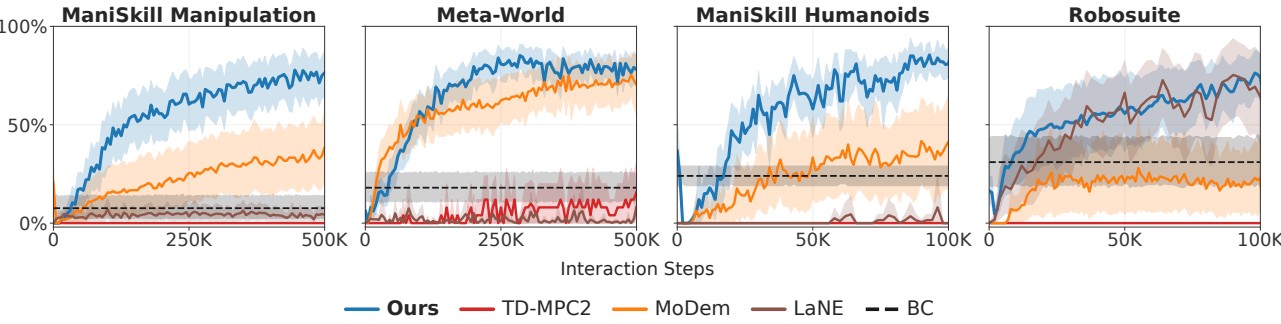

*Figure 5.* **Learning curves**. Success rate as a function of interaction steps for each of the four domains that we consider, averaged across all tasks and 5 random seeds. The shaded area corresponds to a 95% confidence interval. Our method consistently outperforms baselines.

1997; Pomerleau, 1988), $\pi_{BC}$ to collect informative samples during early stages of training. In phase 2, we gradually phase out the (frozen) pre-trained policy and start collecting samples by planning through the world model, which is learned via interactive learning. An overall diagram of our training strategy can be found in Figure 3.

**Phase 1: Pretraining**. One of the main bottlenecks of RL in long-horizon sparse reward tasks is the low-informative data that is collected at the early stages of training. As early data tends to contain no rewards, learning a meaningful representation for such states becomes challenging. Traditional methods tend to start collecting data with a randomly initialized policy that usually struggles to find any rewards in the environment. For this reason, we jointly pre-train a policy $\pi_{BC}$ and an encoder $h_{BC}$ on the full demonstration dataset $\mathcal{D}$ using the classic behavioral cloning (BC) loss as an objective function:

$$\mathcal{L}_{BC}(\theta) = \mathop{\mathbb{E}}_{(\mathbf{o},\mathbf{a})\sim\mathcal{D}} \left[ -\log \pi_\theta(\mathbf{a}|h_\theta(\mathbf{o})) \right], \qquad (6)$$

where the policy learns to imitate the behaviors encoded in the dataset. At interaction time, the interactive policy $\pi_{RL}$ and the world model encoder, h, are initialized with their pre-trained analogs $\pi_{BC}, h_{BC}$.

Given that we focus on datasets with limited demonstrations, behavioral cloning can be prone to overfitting (Peters et al., 2010; Parisi et al., 2022; Duan et al., 2017). To mitigate this, we regularly evaluate $\pi_{BC}$ during pretraining by rolling out episodes in the environment. We use early stopping on the evaluation set (Yao et al., 2007) to select the best-performing policy for the interactive learning phase.

**Phase 2: Interactive Learning**. After initial pretraining of the encoder and policy, the agent starts collecting data from the environment to learn using offline reinforcement learning (RL). In order to utilize the demonstrations, we follow Hansen et al. (2023) by sampling from the replay buffer $\mathcal{B}$ and the demonstration dataset $\mathcal{D}$ at each update step. Specifically, every time we sample a batch, a fraction of the samples come from $\mathcal{D}$ while the remaining fraction comes

*Table 1.* **Experimental setup**. We consider **16** challenging visual manipulation tasks in 4 different domains. Domains that empirically present a slower convergence are given a bigger budget of interactions. The number of stages is determined according to the nature of the task and by the typical horizon of demonstrations.

| Domain | Tasks | Demos | Interactions | Stages |
|---|---|---|---|---|
| **ManiSkill** | 5 | 5-100 | 500k | 3 |
| **Meta-World** | 5 | 5 | 500k | 2 |
| **Robosuite** | 4 | 5-25 | 100k | 1 |
| **Humanoids** | 2 | 5 | 100k | 3 |

from. This approach prevents collected data from quickly outnumbering the demonstrations. While the sampling ratio is a tunable hyperparameter, we empirically found that an initial 50% demonstration ratio works well for most tasks.

Therefore, as detailed in Algorithm 1, at each update step, the world model gets updated as explained in Section 3.1. The agent will then proceed to interact with the environment to collect more data that will be stored in the replay buffer.

Similar to Lancaster et al. (2024), we use annealing to control the probability of a sample coming from $\pi_{BC}$ or from the planning module of the world model. In this way, the data distribution of the replay buffer $\mathcal{B}$ is initially biased toward the one of the dataset $\mathcal{D}$. This technique aims to collect more informative data than the one a purely random policy would collect during early stages of training. As the world model starts learning and $\pi_{RL}$ is able to collect more meaningful samples, we increase the annealing coefficient $\alpha_t$ to improve the diversity of collected samples and stop relying on the suboptimal pre-trained policy $\pi_{BC}$. Eventually, $\alpha_t$ will converge to 1, at which point all samples will come from planning with $\pi_{RL}$ (see Algorithm 1).

## 4. Experiments

We consider **16** challenging visual multi-stage manipulation tasks with a long-horizon for our experimental evaluation. This includes **5** manipulation tasks from *ManiSkill3* (Tao

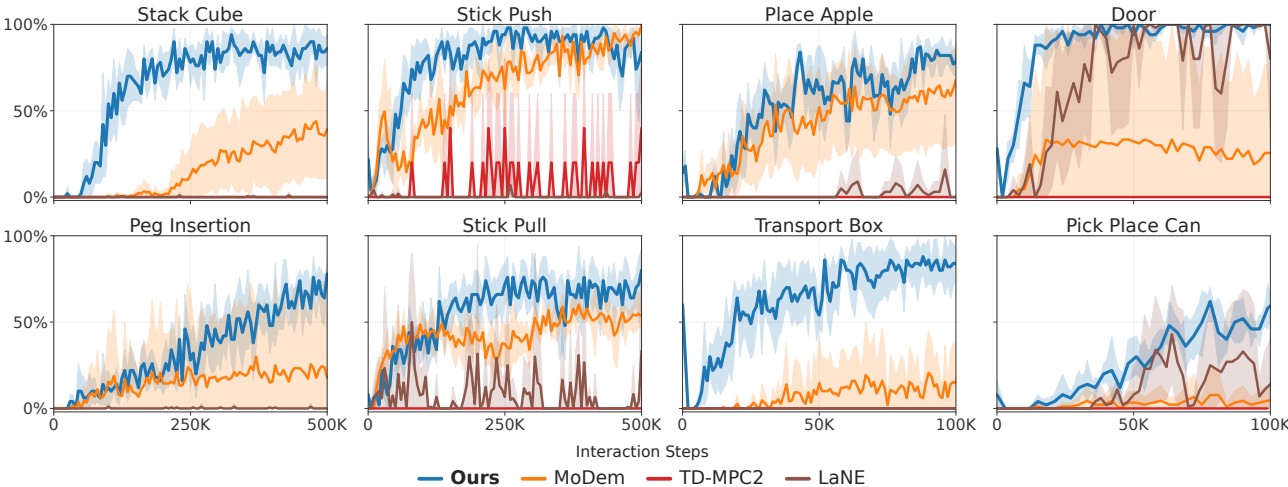

*Figure 6.* **Challenging tasks**. Success rate of our method and baselines on the 2 hardest tasks from each domain. Averaged across 5 random seeds. LaNE results in Robosuite are kindly provided by Zhao et al. (2024). Shaded areas correspond to 95% confidence intervals.

et al., 2024b), **5** manipulation tasks from *Meta-World* (Yu et al., 2021) and **4** manipulation tasks from *Robosuite* (Zhu et al., 2022). Additionally, we include **2** humanoid manipulation tasks from *ManiSkill3* with a high-dimensional action space, which we refer to as *ManiSkill Humanoids* in our evaluations. We place a strong emphasis on long-horizon precise manipulation, which is why we select the most challenging tasks from each domain. We relate the difficulty to the required precision of a task, its horizon, and the level of randomization in the scene (see Appendix D.3 for further details on difficulty categorization). For each task, the agent is given a constrained budget of demonstrations and interaction steps (see Table 1). To allow most baselines to solve the task, we set the interaction budget and demonstrations to a different amount for each task. For a complete list of details on our experimental setup, please refer to Table 1 and Appendix D. Through our evaluations, we aim to answer the following questions:

1. Can our proposed method effectively accelerate MBRL with demonstrations in long-horizon multi-stage tasks?

2. What is the relative importance of each algorithmic component of Demonstration-Augmented Reward, Policy, and World Model Learning (DEMO³), and how does it scale with the amount of demonstration data?

3. How do sparse reward functions compare to our learned rewards at different levels of stage granularity?

### 4.1. Baselines

To assess our method's effectiveness, we compare it against three relevant approaches. A complete comparison of other methods can be found in Appendix B.

**MoDem** (Hansen et al., 2023) is a MBRL algorithm designed to enhance data-efficiency in visual control tasks

with sparse rewards. Similarly to our method, MoDem employs a three-phase framework: policy pretraining, seeding, and interactive learning with oversampling of demonstration data. The authors show state-of-the-art performance on Meta-World and Adroit domains from visual inputs and sparse rewards.

**LaNE** (Zhao et al., 2024) is a data-efficient model-free RL method for sparse-reward tasks from visual inputs. LaNE utilizes a pre-trained feature extractor to learn an embedding space and **rewards the agent for exploring regions near the demonstrations within this latent space**. The authors also show state-of-the-art data-efficiency in the Robosuite environment with a limited amount of demonstrations.

**TD-MPC2** (Hansen et al., 2022; 2024) is the state-of-the art MBRL algorithm for control tasks. It combines temporal difference learning with model predictive control (MPC) and constitutes the backbone of our approach. Compared to TD-MPC, TD-MPC2 includes a series of algorithmic changes that improve robustness and scaling.

### 4.2. Benchmark Results

The main result of our evaluations (see Figure 5) compares the data-efficiency of our method against the proposed baselines. On average, our method achieves 40% better data-efficiency than the proposed baselines. Notably, in *ManiSkill3*, our most difficult domain, our method averages a 75% success rate after only 500k steps, performing 50% better than the second-best baseline. Furthermore, our DEMO³ can deal better with the high-dimensional action space in ManiSkill Humanoids. Interestingly, while our DEMO³ does better on average, it thrives in the most difficult tasks where the horizon and precision of the task are the highest. While the results of LaNE in the Robosuite domain are cer-

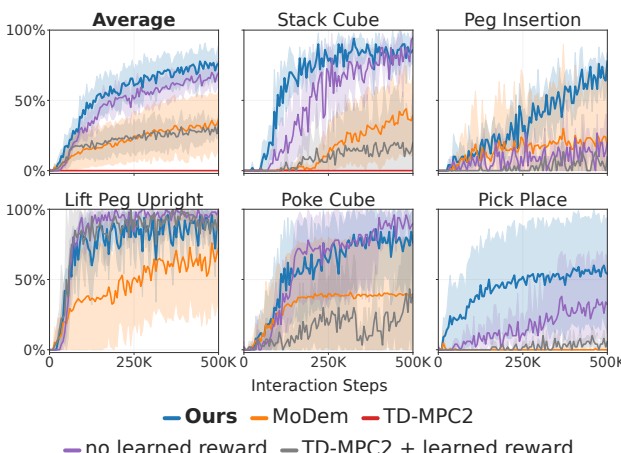

*Figure 7.* **Ablations**. Success rate as a function of interaction steps for variations of our method on all 5 ManiSkill manipulation tasks. Averaged across 5 random seeds. Baselines included for completeness. Shaded areas correspond to 95% confidence intervals.

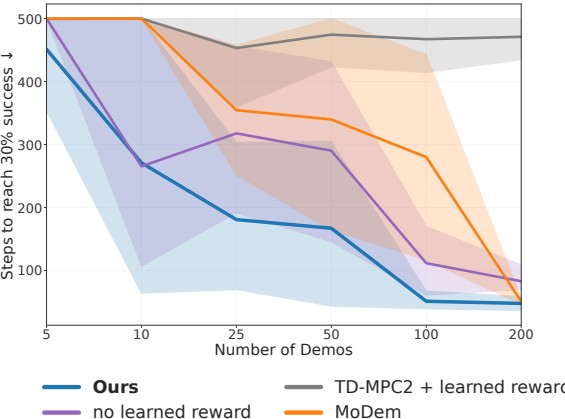

*Figure 8.* **Demonstration efficiency**. Number of steps to reach a critical success rate (30%) as a function of demonstration count. Data points that did not converge are assigned a 500k step count. Results are aggregated over 2 challenging manipulation tasks (Stack Cube and Peg Insertion) and averaged across 5 seeds. Shaded areas correspond to a 95% confidence interval.

tainly impressive, the same approach doesn't transfer to the rest of the domains.

Figure 6 shows the learning curves for the most challenging task of each benchmark. Particularly, the ManiSkill tasks, Peg Insertion, and Stack Cube, require a very high level of precision and a long horizon. As shown in 6, DEMO³ is the only algorithm to reliably solve both tasks in the interaction budget. While performance is quite matched with LaNE (Zhao et al., 2024) in Robosuite, LaNE uses a pre-trained encoder to preprocess image observations while our method is completely learned from scratch. Finally, TD-MPC2 struggles to get any performance as is typical for pure RL algorithms learning from sparse rewards. Overall, DEMO³ shows the highest degree of robustness and efficiency on the proposed long-horizon tasks.

### 4.3. Analysis

**Relative importance of each component** . Figure 7 shows the effect of removing dense reward learning, policy pretraining, and demonstration oversampling in the 5 manipulation tasks from *ManiSkill3*. Interestingly, a considerable jump in performance is brought by pretraining and oversampling from the demonstration dataset with the TD-MPC2 backbone (*no learned reward*). The effect of reward learning becomes evident in long-horizon tasks where advancing stages can be very challenging without rewards (e.g., Peg Insertion, Pick Place), especially when reducing the number of demonstrations (see Appendix A.2).

**Demonstration efficiency** In Figure 8, we experiment with different dataset sizes and evaluate the data-efficiency of different baselines using demonstrations. While most methods scale well with the number of demonstrations,

DEMO³ shows the strongest performance in the lowest regime of demonstrations. This is particularly evident in challenging tasks such as Peg Insertion and Stack Cube, where DEMO³ is the only method capable of reaching meaningful performance within the interaction budget (see Appendix A.2) with only 5 demonstrations.

**Wall-time comparison** Our method achieves competitive wall-time performance, ranking as the second fastest among all evaluated algorithms (Table 2). While slightly slower than TD-MPC2, we attribute the overhead to the additional computation required for reward learning. Importantly, DEMO³ performs much faster than other demonstration-augmented RL approaches, such as Modem and LaNE.

*Table 2.* **Wall-time.** Hours per 100k interaction steps, averaged across 5 seeds and all tasks in Robosuite. Lower is better ↓.

| Algorithm | Time (hours) ↓ |
|-----------|----------------|
| LaNE      | 20.40          |
| MoDem     | 8.37           |
| TD-MPC2   | **4.84**       |
| Ours      | 5.19           |

**Reward granularity** Figure 9 illustrates the data-efficiency of our method across the same tasks under varying reward granularities: 1 stage, 2 stages, 3 stages, and dense rewards. Please refer to D for details about the stage definitions. As expected, dense rewards provide the optimal learning signal, resulting in the fastest task completion. Remarkably, our method achieves near-equal performance with only 2 stages, demonstrating that stage indicators combined

with a minimal set of demonstrations are sufficient to guide the learning process effectively. This result shows that over-engineered dense reward functions can easily be avoided by only providing a small number of demonstrations. By relying solely on stage indicators and demonstrations, our method simplifies the reward design process while maintaining high performance in complex long-horizon tasks.

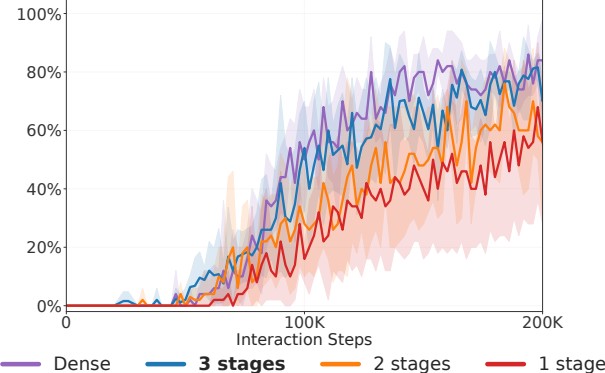

*Figure 9.* **Reward granularity**. Success rate of our method with increasing granularity in the stage division of a task. Results are aggregated over 2 challenging manipulation tasks (Stack Cube and Peg Insertion) and averaged across 5 seeds. The shaded area corresponds to a 95% confidence interval.

## 5. Related work

**Model-based RL**   Model-based Reinforcement Learning (MBRL) improves data-efficiency by leveraging a model of the environment to guide decision-making. These models can be either prior-based, such as physics-based simulators, or learned, where the agent approximates a dynamic model of the world from data. World models (Schmidhuber, 1990; Ha & Schmidhuber, 2018) are an internal representation of the environment that enables planning and policy learning without direct interaction. A notable example is MuZero (Schrittwieser et al., 2020), which extends value-based planning by implicitly learning environment dynamics. Recent advances, such as Dreamer (Hafner et al., 2020; 2022; 2024) and TD-MPC (Hansen et al., 2022; 2024), improve learning in high-dimensional spaces, allowing MBRL to scale to complex visual and continuous control tasks.

**Demonstration-Augmented RL**   Learning policies purely through trial and error can be inefficient and unstable, prompting research into leveraging demonstrations to enhance RL. During online interactions, demonstrations can serve as off-policy experience (Hester et al., 2018; Kapturowski et al., 2018; Ball et al., 2023; Nair et al., 2018; Escontrela et al., 2022) or for on-policy regularization (Kang et al., 2018; Rajeswaran et al., 2017). Alternatively, demonstrations can be used to estimate reward functions for RL (Xie et al., 2018; Aytar et al., 2018; Vecerik et al., 2019; Zolna

et al., 2020; Singh et al., 2019). *In this work, we leverage demonstrations in multiple ways simultaneously: learning an initial policy, the world model, and a reward function.*

**Reward Learning**   Designing rewards is challenging due to the need for extensive domain knowledge, prompting the development of data-driven reward learning methods. Rewards can be learned from offline datasets by classifying goals (Smith et al., 2019; Kalashnikov et al., 2021; Du et al., 2023) or estimating goal distances (Zakka et al., 2022). Alternatively, inverse RL approaches (Ng et al., 2000; Ziebart et al., 2008; Ho & Ermon, 2016; Fu et al., 2017) leverage online interactions to infer a reward function from expert demonstrations. Additionally, reward shaping techniques (Trott et al., 2019; Wu et al., 2021; Memarian et al., 2021; Escontrela et al., 2022) transform sparse rewards into dense rewards using specific domain knowledge. The reward learning in this work builds upon DrS (Mu et al., 2024), with modifications tailored for compatibility with MBRL.

## 6. Conclusions and Future Directions

In this work, we tackle the challenge of learning long-horizon manipulation skills with sparse rewards using only proprioceptive and visual feedback. We propose DEMO[3], a demonstration-augmented MBRL algorithm that simultaneously learns a reward, policy and world-model for multi-stage manipulation. Our experiments (Section 4) show that our method achieves 40% better performance than the current state-of-the-art. Additionally, DEMO[3] excels at the most difficult tasks, converging up to 4x faster than current methods. We propose possible future research directions:

Firstly, although our experiments show that a small number of demonstrations is sufficient for accelerated learning, we do not explore the effect of using different sources, *e.g.*, data collected via human teleoperation. While prior work suggests such sources have limited impact on RL performance (Hansen et al., 2023), improving robustness to diverse demonstration types remains relevant for deployment.

Another limitation is that our experiments are limited to simulation. We aim to deploy DEMO[3] on real robot hardware to evaluate its generalization under domain shift and whether the same constrained regime of demonstrations and interaction samples can yield reliable real-world policies. Notably, Hansen et al. (2023) shows that a similar pipeline transfers well to physical systems. Since DEMO[3] improves on these components in simulation, we are optimistic its benefits will carry over.

Finally, while our 50% demonstration sampling ratio yields strong results, more sophisticated strategies (Schaul et al., 2016) could further enhance learning efficiency.

## Acknowledgements

Nicklas Hansen is supported by NVIDIA Graduate Fellowship. Stone Tao is supported in part by the NSF Graduate Research Fellowship Program grant under grant No. DGE2038238.

## Impact Statement

This paper presents work whose goal is to advance the field of robot learning. While there are many potential societal consequences in developing embodied intelligence for the real world, we do not feel our work presents any particular implications that should be highlighted here.

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

# A. Additional Results

## A.1. Single Task Experimental Results

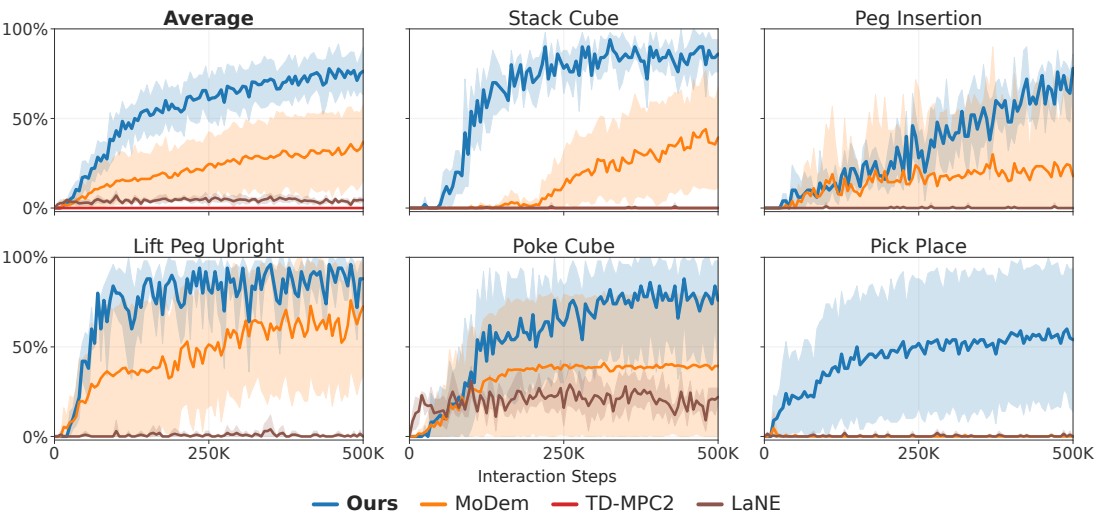

*Figure 10.* **ManiSkill Manipulation results**. Results averaged across 5 seeds. The shaded area corresponds to a 95% confidence interval.

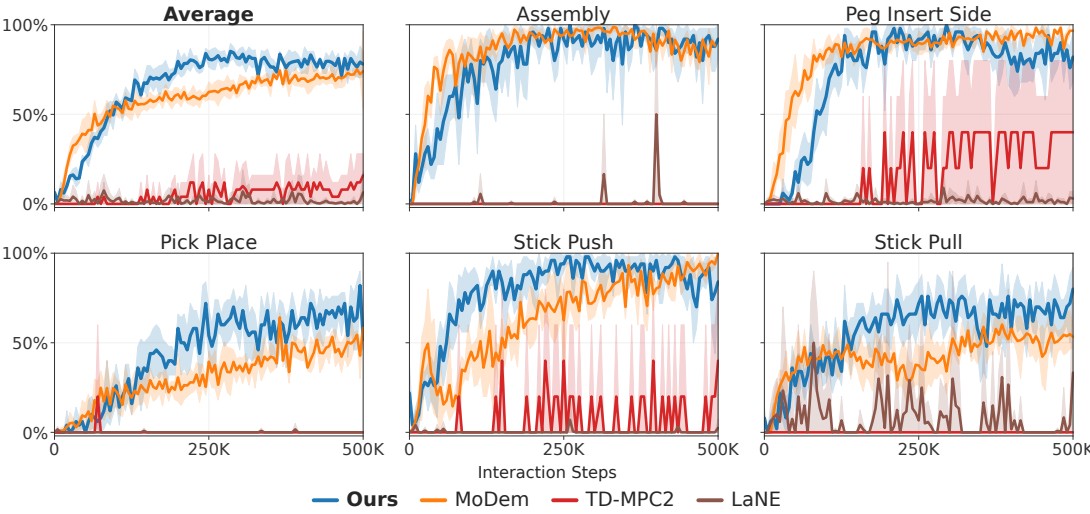

*Figure 11.* **Meta-World results**. Results averaged across 5 seeds. The shaded area corresponds to a 95% confidence interval.

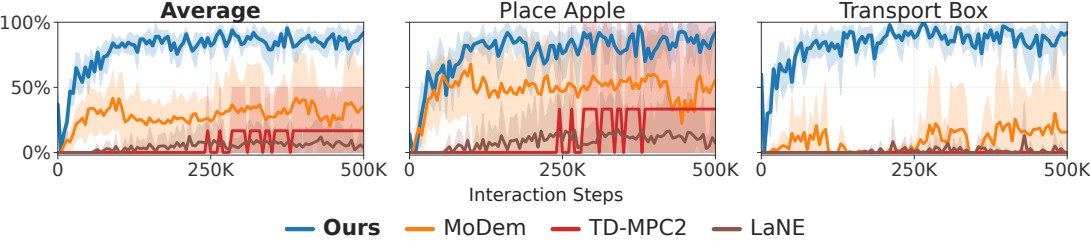

*Figure 12.* **ManiSkill Humanoids results**. Results averaged across 5 seeds. The shaded area corresponds to a 95% confidence interval.

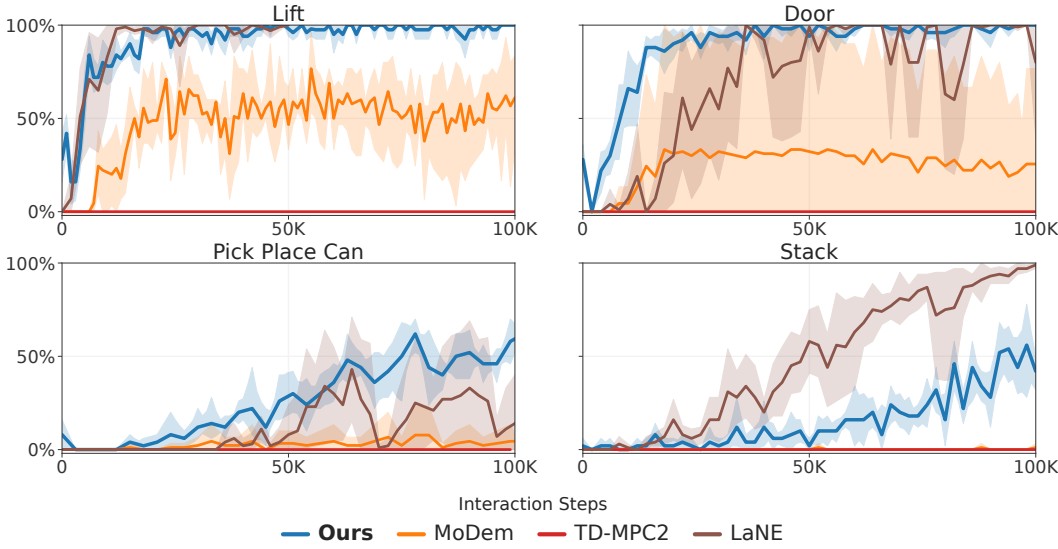

*Figure 13.* **Robosuite results**. Results averaged across 5 seeds. The shaded area corresponds to a 95% confidence interval.

## A.2. Additional Ablations

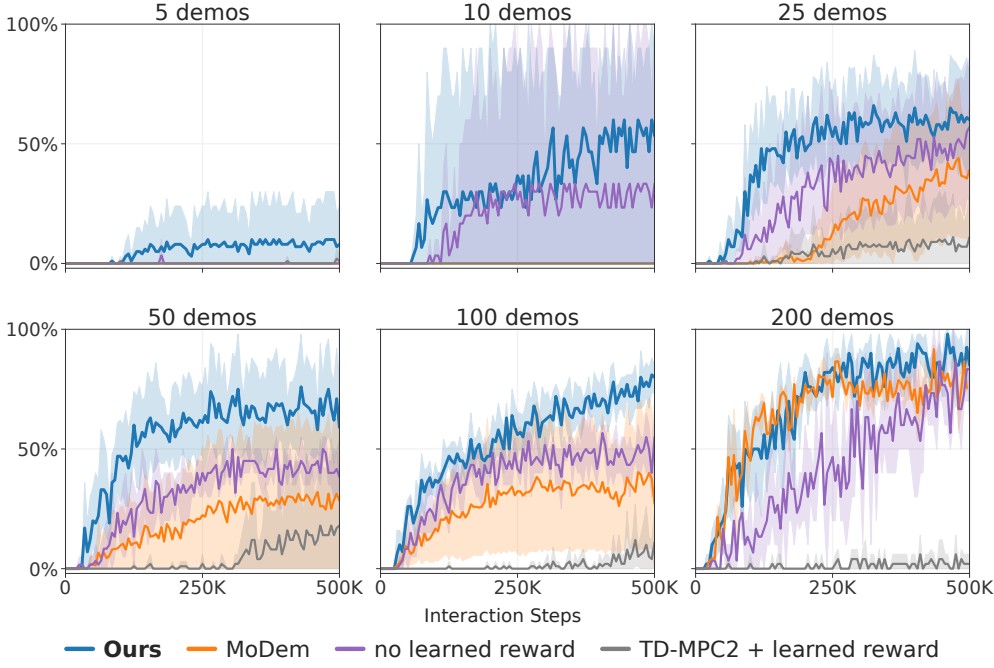

*Figure 14.* **Demonstration ablation**. Success rate on an increasing number of demonstrations (5-200) in the 2 most challenging manipulation tasks in ManiSkill (Stack Cube and Peg Insertion). DEMO[3] is the only method that has a relative success with only 5 demonstrations. Results are aggregated over both tasks and averaged across 5 seeds.

## A.3. Learned Reward

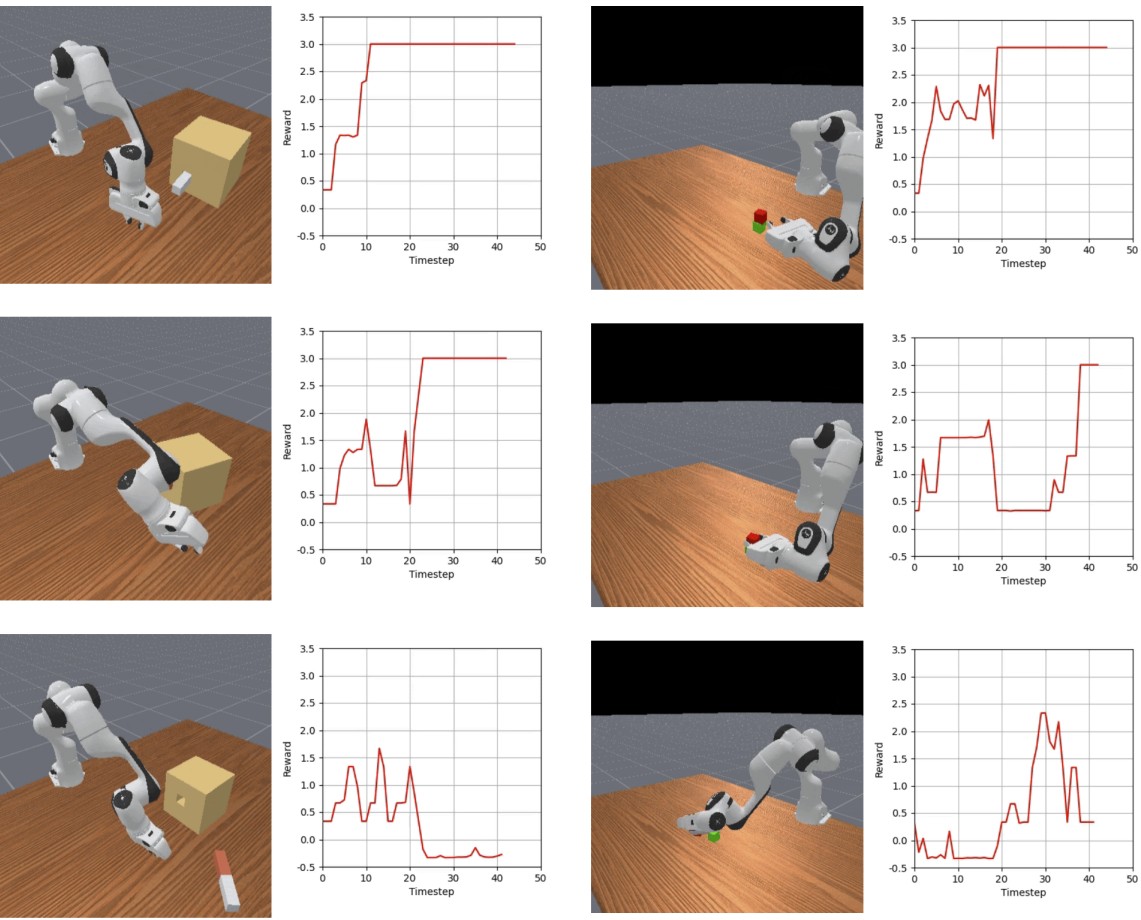

*Figure 15.* **Learned reward animation**. Visualization of the learned dense reward across three representative rollouts—successful, failed, and semi-successful. The reward curves closely track task progress.

# B. Baselines

## B.1. Comparison to prior work

In Table 3, we compare key components of DEMO[3] to relevant prior methods on demonstration-augmented RL. Our approach is the only one incorporating online reward learning in multi-stage settings for visual inputs and sparse rewards.

*Table 3.* **Comparison to prior work**. We compare DEMO[3] to relevant approaches and ablations. Selected baselines are highlighted.

| Method | Visual Inputs | Sparse Rewards | Multi-Stage | Online Reward Learning |
|:---:|:---:|:---:|:---:|:---:|
| **Ours** | ✓ | ✓ | ✓ | ✓ |
| LaNE (Zhao et al., 2024) | ✓ | ✓ | ✗ | ✓ |
| MoDem (Hansen et al., 2023) | ✓ | ✓ | ✗ | ✗ |
| CoDER (Zhan et al., 2022) | ✓ | ✓ | ✗ | ✗ |
| SAC + DrS (Mu et al., 2024) | ✗ | ✓ | ✓ | ✗ |
| AMP (Escontrela et al., 2022) | ✗ | ✗ | ✗ | ✓ |

## B.2. Baseline Implementations

**TD-MPC2** We use the official implementation[1] with default parameters. We add two extra layers to the convolutional encoders to handle the higher image resolution of the Meta-World and Robosuite benchmarks.

**MoDem** We use the official implementation[2] with default parameters. To process observations containing multiple images, we add an extra encoder to the world model and average all the embeddings.

**LaNE** We use the code from the official implementation[3] with default parameters. To adapt it to our experimental setup, we add extra encoders to handle the additional observations. The proprioceptive state is passed through an MLP, and its embedding is averaged with the other inputs. Additionally, we adapt the algorithm to handle infinite-horizon MDPs by removing value function bootstrapping from the MDP.

---

[1] https://github.com/nicklashansen/tdmpc2
[2] https://github.com/facebookresearch/modem
[3] https://github.com/PhilipZRH/LaNE

# C. Demonstrations

All of our demonstrations are obtained by training a TD-MPC2 model with dense rewards and state observations. The model trained on state observations is then used to rollout $N$ episodes in the stage-based environment, from where we query image observations, proprioceptive states, and sparse stage rewards. Please find below a detailed table on the number of demonstrations used per task.

*Table 4.* **Number of demonstrations for each task**. We use the minimum amount of demonstrations (empirically determined) to ensure that the best-performing algorithm can solve the task in the given interaction budget.

| Domain | Task | Number of Demonstrations |
|---|---|---|
| ManiSkill Manipulation | **Peg Insertion** | 100 |
| | **Pick Place** | 100 |
| | **Stack Cube** | 25 |
| | **Poke Cube** | 5 |
| | **Lift Peg Upright** | 5 |
| Meta-World | **Assembly** | 5 |
| | **Peg Insert Side** | 5 |
| | **Stick Push** | 5 |
| | **Stick Pull** | 5 |
| | **Pick Place** | 5 |
| ManiSkill Humanoids | **Place Apple** | 5 |
| | **Transport Box** | 5 |
| Robosuite | **Lift** | 5 |
| | **Door** | 10 |
| | **Pick Place Can** | 10 |
| | **Stack Blocks** | 20 |

# D. Experiment Details

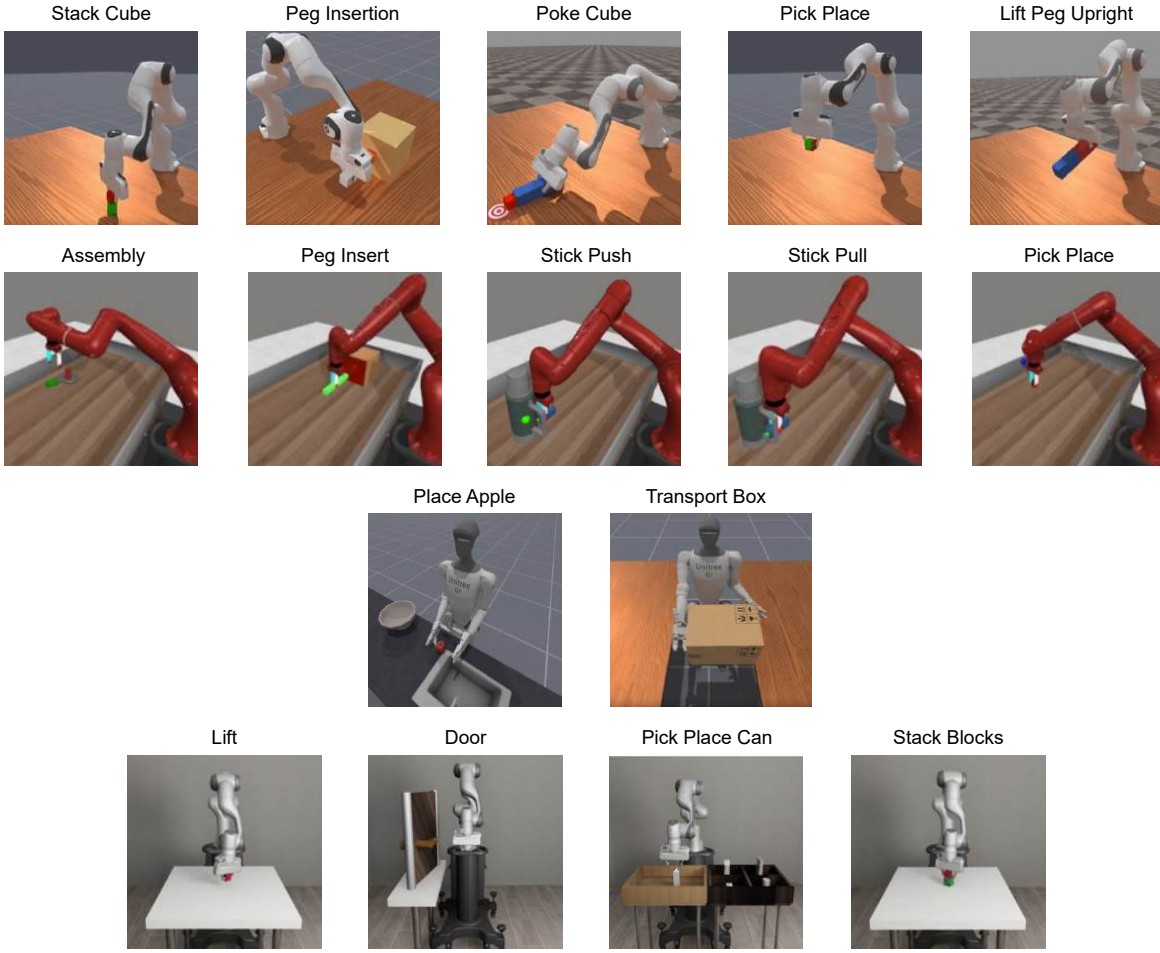

*Figure 16.* **All tasks.** Visual description of all tasks organized by domains. In descending order: ManiSkill Manipulation, Meta-World, ManiSkill Humanoids, and Robosuite.

## D.1. Domain Implementation

*Table 5.* **Implementation details for each of our four domains.** Time horizon is measured in agent steps (policy forward passes). *Proprio.* stands for Proprioceptive State. Each domain uses different image resolutions according to the detail of the scene.

|  | **ManiSkill Manipulation** | **Meta-World** | **ManiSkill Humanoids** | **Robosuite** |
|---|---|---|---|---|
| **Time Horizon** | 100 | 100 | 100 | 100 |
| **Image Size** | $128 \times 128$ | $224 \times 224$ | $128 \times 128$ | $128 \times 128$ |
| **Observations** | RGB(x2) + Proprio. | RGB + Proprio. | RGB(x2) + Proprio. | RGB(x2) |
| **Cameras** | Hand + Front | Front | Head + Front | Hand + Front |
| **Action Repeat** | 2 | 2 | 2 | 1 |
| **Action Dim** | 7 | 4 | 25 | 7 |

## D.2. Stage Definitions

*Table 6.* **ManiSkill Manipulation stage definitions**.

| Task | Stage 1 | Stage 2 | Success Criteria |
|---|---|---|---|
| **Stack Cube** | Cube A grabbed. | Cube A above Cube B. | Cube A stacked on top of Cube B. |
| **Peg Insertion** | Peg grabbed. | Peg aligned with hole. | Peg inserted in hole. |
| **Pick Place** | Cube grabbed. | Cube close to goal. | Cube at goal. |
| **Poke Cube** | Peg grabbed. | Peg touching Cube. | Cube at goal. |
| **Lift Peg Upright** | Peg grabbed. | Peg upright. | Peg upright on desk. |

*Table 7.* **Meta-World stage definitions**.

| Task | Stage 1 | Success Criteria |
|---|---|---|
| **Assembly** | Grab hook. | Pass nut through pole. |
| **Peg Insert** | Peg grabbed. | Peg inserted in hole. |
| **Stick Push** | Stick grabbed. | Object pushed to goal location. |
| **Stick Pull** | Stick grabbed. | Object pulled to goal location. |
| **Pick Place** | Cube grabbed. | Cube is static at goal. |

*Table 8.* **ManiSkill Humanoids stage definitions**.

| Task | Stage 1 | Stage 2 | Success Criteria |
|---|---|---|---|
| **Place Apple** | Apple is grabbed. | Apple is above bowl. | Apple is inside bowl. |
| **Transport Box** | Box grabbed with 2 hands. | Box above table 2. | Box is on table 2. |

*Table 9.* **Robosuite stage definitions**.

| Task | Success Criteria |
|---|---|
| **Lift** | Block lifted above the desk. |
| **Door** | Door is open. |
| **Pick Place Can** | Can is at goal location. |
| **Stack** | Block A is in contact with Block B and above the ground. |

## D.3. Difficulty Categorization

Across this paper, we often refer to some tasks as more *difficult* than others. To characterize task difficulty, we follow (Tao et al., 2024a). As in previous work on demonstration-augmented reinforcement learning (RL), we observe that environments with high complexity and substantial initial state randomization are typically more difficult to solve and require a larger number of demonstrations. For example, the *Peg Insertion* task from *ManiSkill* exhibits significant variability in the peg's position, orientation, and the hole's size. Consequently, around 100 demonstrations are needed to solve the task from visual inputs. In contrast, a task like *Meta-World Assembly* requires only 5 demonstrations to be successfully solved. Figure 17 qualitatively compares the different initial states of these two tasks. We hypothesize that this effect is related to the distributional coverage of the demonstration dataset: higher randomization reduces the likelihood that the agent encounters familiar states during training if the dataset is limited. Therefore, as the variability in a task's initial state increases, this variability must also be well-represented in the dataset to ensure effective learning.

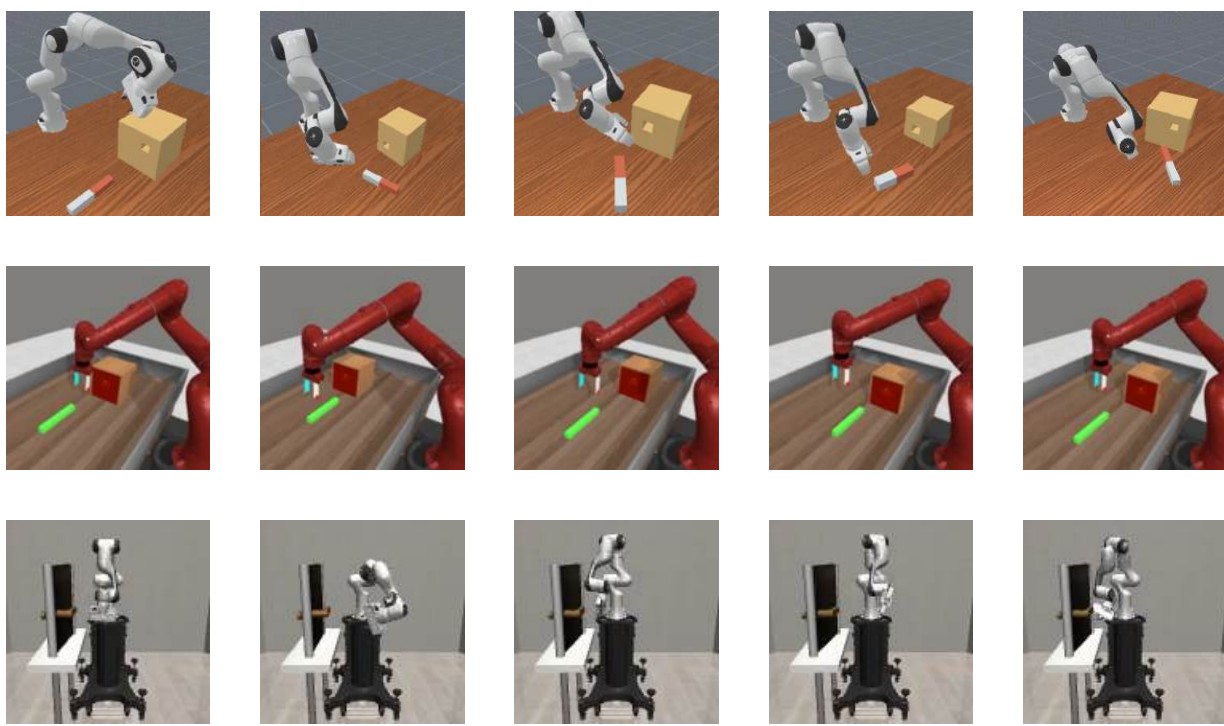

*Figure 17.* **Randomization comparison**. Qualitative comparison of both Peg Insertion tasks in the ManiSkill and Meta-World domain and Door task in Robosuite. Visibly, ManiSkill presents the highest level of randomization, not only varying the initial state at reset but also changing the geometric properties of the objects.

# E. Implementation Details

## E.1. Model Architecture

Following TD-MPC2, all modules are implemented as MLPs. Here, as an example, we summarize our architecture for a single-camera Meta-World task using PyTorch-like notation:

```
Architecture: TD-MPC2 World Model
Encoder: ModuleDict(
  (rgb_frontview): Sequential(
    (0): ShiftAug()
    (1): PixelPreprocess()
    (2): Conv2d(3, 32, kernel_size=(7, 7), stride=(2, 2))
    (3): ReLU(inplace=True)
    (4): Conv2d(32, 32, kernel_size=(5, 5), stride=(2, 2))
    (5): ReLU(inplace=True)
    (6): Conv2d(32, 32, kernel_size=(3, 3), stride=(2, 2))
    (7): ReLU(inplace=True)
    (8): Conv2d(32, 32, kernel_size=(3, 3), stride=(2, 2))
    (9): ReLU(inplace=True)
    (10): Conv2d(32, 32, kernel_size=(3, 3), stride=(1, 1))
    (11): Flatten(start_dim=1, end_dim=-1)
    (12): Linear(in_features=512, out_features=512, bias=True)
    (13): SimNorm(dim=8)
  )
)
Dynamics: Sequential(
  (0): NormedLinear(in_features=519, out_features=512, bias=True, act=Mish)
  (1): NormedLinear(in_features=512, out_features=512, bias=True, act=Mish)
  (2): NormedLinear(in_features=512, out_features=512, bias=True, act=SimNorm)
)
Reward: Sequential(
  (0): NormedLinear(in_features=519, out_features=512, bias=True, act=Mish)
  (1): NormedLinear(in_features=512, out_features=512, bias=True, act=Mish)
  (2): Linear(in_features=512, out_features=101, bias=True)
)
Policy prior: Sequential(
  (0): NormedLinear(in_features=512, out_features=512, bias=True, act=Mish)
  (1): NormedLinear(in_features=512, out_features=512, bias=True, act=Mish)
  (2): Linear(in_features=512, out_features=14, bias=True)
)
Q-functions: Vectorized [Sequential(
  (0): NormedLinear(in_features=519, out_features=512, bias=True, dropout=0.01, act=Mish)
  (1): NormedLinear(in_features=512, out_features=512, bias=True, act=Mish)
  (2): Linear(in_features=512, out_features=101, bias=True)
), Sequential(
  (0): NormedLinear(in_features=519, out_features=512, bias=True, dropout=0.01, act=Mish)
  (1): NormedLinear(in_features=512, out_features=512, bias=True, act=Mish)
  (2): Linear(in_features=512, out_features=101, bias=True)
), Sequential(
  (0): NormedLinear(in_features=519, out_features=512, bias=True, dropout=0.01, act=Mish)
  (1): NormedLinear(in_features=512, out_features=512, bias=True, act=Mish)
  (2): Linear(in_features=512, out_features=101, bias=True)
)]
Learnable parameters: 5,448,748
Discriminator Architecture: Discriminator(
  (nets): ModuleList(
    (0): Sequential(
      (0): Linear(in_features=512, out_features=32, bias=True)
      (1): Sigmoid()
      (2): Linear(in_features=32, out_features=1, bias=True)
    )
  )
```

## E.2. Hyperparameters

Most of the hyperparameters remain unchanged from our backbone algorithm, TD-MPC2. Here, we list some of the most relevant to our method and highlight the ones that are unique to our approach. Please refer to the TD-MPC2 paper (Hansen et al., 2024) for a complete list of hyperparameters.

*Table 10.* Hyperparameters used in the training setup.

| Hyperparameter | Value |
| --- | --- |
| **Replay buffer** | |
| Capacity | $300,000$ |
| Sampling | Uniform |
| **Architecture (5M)** | |
| Encoder arch. | ConvNet (image inputs) |
| | MLP (state inputs) |
| Conv. layers | 7 (Meta-World) |
| | 5 (Otherwise) |
| Encoder MLP dim | 256 |
| Dynamics MLP dim | 512 |
| Latent state dim | 512 |
| Task embedding dim | 96 |
| **Optimization** | |
| Update-to-data ratio | 1 |
| Batch size | 256 |
| Joint-embedding coef. | 20 |
| Reward prediction coef. | 0.1 |
| Value prediction coef. | 0.1 |
| Temporal coef. ($\lambda$) | 0.5 |
| $Q$-fn. momentum coef. | 0.99 |
| Policy prior entropy coef. | $1 \times 10^{-4}$ |
| Policy prior loss norm. | Moving ($5\%$, $95\%$) percentiles |
| Optimizer | Adam |
| Learning rate | $3 \times 10^{-4}$ |
| Encoder learning rate | $1 \times 10^{-4}$ |
| **Pretraining** | |
| Pretraining loss | Behavioral cloning |
| BC Policy Architecture | MLP |
| MLP dim | 512 |
| Optimizer | Adam |
| Learning rate | $3 \times 10^{-4}$ |
| Encoder learning rate | $3 \times 10^{-4}$ |
| $\alpha_0$ | $5 \times 10^{-5}$ |
| **Reward learning** | |
| Discriminator architecture | MLP |
| MLP dim | 32 |
| Discriminator learning rate | $3 \times 10^{-4}$ |
| Discriminator optimizer | Adam |
| Batch size | 256 |
| $\beta$ | 1/3 |
| Demo. sampling ratio | $50\%$ |

**E.3. Computational Resources**

All our experiments run on a single NVIDIA GeForce RTX 3090 GPU and 32GB of RAM to store collected samples.

