# OpenReview forum: "Multi-Stage Manipulation with Demonstration-Augmented Reward, Policy, and World Model Learning"
_ICML.cc/2025/Conference — ICML 2025 poster_

### Official Review · Reviewer_di7L · 2025-03-09

**Overall Recommendation:** 3

**Summary:**

This paper introduces DEMO3 (Demonstration-Augmented Reward, Policy, and World Model Learning), a novel framework for solving long-horizon, multi-stage manipulation tasks with sparse rewards. The authors address the challenge of designing dense reward functions and effectively exploring large state-action spaces by leveraging a small number of demonstrations for three key purposes: learning a policy, a world model, and a dense reward function. The approach incorporates multi-stage dense reward learning, a bi-phasic training scheme, and world model learning into a demonstration-augmented reinforcement learning framework. The method is evaluated across 16 sparse-reward tasks spanning four domains, including challenging humanoid visual control tasks, demonstrating improved data-efficiency by an average of 40% and by 70% on particularly difficult tasks compared to state-of-the-art approaches.

**Claims And Evidence:**

The claims made in the submission are well-supported by evidence. The authors claim that their method improves data-efficiency compared to state-of-the-art approaches, which is substantiated by comprehensive experimental results across 16 tasks in four domains. The learning curves in Figure 5 clearly demonstrate that DEMO3 consistently outperforms baseline methods (TD-MPC2, MoDem, and LaNE) in terms of success rate as a function of interaction steps. The claim that the method is particularly effective on difficult tasks is supported by the 70% improvement in performance on complex tasks. The authors also claim that their approach requires only a small number of demonstrations (as few as five), which is validated by their experimental setup described in Table 1.

**Essential References Not Discussed:**

N/A

**Experimental Designs Or Analyses:**

The experimental design is robust and comprehensive. The authors evaluate their method on 16 tasks across four domains, with varying levels of complexity and different numbers of stages. The experiments use 5 random seeds to ensure statistical significance, and the results are presented with 95% confidence intervals. The learning curves in Figure 5 provide a clear visualization of the method's performance over time, and the summary of results in Figure 1 offers a concise comparison with baselines. The ablation studies help to isolate the contributions of different components of the framework. The authors also provide details on the number of demonstrations used for each domain and the interaction budget, which helps to contextualize the results.

**Methods And Evaluation Criteria:**

The proposed methods and evaluation criteria are appropriate for the problem at hand. The authors use success rate as the primary evaluation metric, which is a clear and relevant measure for manipulation tasks. The experimental setup includes a diverse set of 16 tasks across four domains (ManiSkill Manipulation, Meta-World, Robosuite, and ManiSkill Humanoids), providing a comprehensive evaluation of the method's generalizability. The comparison against strong baselines (TD-MPC2, MoDem, and LaNE) ensures a fair assessment of the method's performance. The authors also conduct ablation studies to analyze the relative importance of each component of their framework, which helps to validate the design choices.

**Other Comments Or Suggestions:**

- The paper would benefit from a more detailed discussion of the limitations of the approach and potential directions for future work.
- A more thorough analysis of the computational requirements of the method would be valuable, particularly for real-time applications.
- It would be interesting to see how the method performs with varying numbers of demonstrations, to better understand the trade-off between demonstration quantity and performance.
- The paper could discuss more explicitly how the approach might be adapted for real-world robotic systems, addressing challenges such as sensor noise and actuation delays.

**Other Strengths And Weaknesses:**

Strengths:
- The paper addresses a significant challenge in reinforcement learning: solving long-horizon, multi-stage manipulation tasks with sparse 2. rewards.
- The proposed method is data-efficient, requiring only a small number of demonstrations (as few as five).
- The approach is evaluated on a diverse set of tasks, demonstrating its generalizability.
- The bi-phasic training scheme is a clever way to leverage demonstrations for both initialization and online learning.

Weaknesses:
- The paper does not extensively discuss the limitations of the approach or potential failure cases.
- While the method is shown to work with as few as five demonstrations, it's not clear how the performance scales with even fewer demonstrations or how it compares to methods that don't use demonstrations at all.
- The computational complexity of the approach is not thoroughly discussed, which is important for practical applications.
- The paper focuses primarily on simulation environments, and it's not clear how well the approach would transfer to real-world robotic systems.

**Questions For Authors:**

- How does the performance of DEMO3 scale with the number of demonstrations? Is there a minimum number of demonstrations required for the method to work effectively, and how does the performance improve with additional demonstrations?
- The paper focuses on simulation environments. Have you explored how the approach might transfer to real-world robotic systems, and what additional challenges might arise in that context?
- The method relies on stage indicators for multi-stage tasks. How sensitive is the approach to the definition of these stages, and how might it be extended to tasks where the stage structure is less clear or where the number of stages might vary between demonstrations?

**Relation To Broader Scientific Literature:**

The authors acknowledge prior work on model-based RL (Ha & Schmidhuber, 2018; Zhang et al., 2018; Kidambi et al., 2020; Hafner et al., 2020; Yu et al., 2020; Hansen et al., 2022; 2024; Sferrazza et al., 2024) and learning from demonstrations (Zhan et al., 2022; Hansen et al., 2023; Lancaster et al., 2024). They also discuss the limitations of existing approaches, such as the challenges of designing dense reward functions and the exploration problems in sparse reward settings. The paper builds upon TD-MPC2 (Hansen et al., 2022; 2024) as its backbone for model-based RL, and draws inspiration from MoDem (Hansen et al., 2023) for its bi-phase training scheme, clearly acknowledging these influences.

**Theoretical Claims:**

The paper makes several theoretical claims about the benefits of their approach, particularly regarding the use of stage-specific discriminators for dense reward learning and the bi-phasic training scheme. These claims are well-founded in reinforcement learning theory, particularly in the context of model-based RL and learning from demonstrations. The authors provide a clear mathematical formulation of their approach, including the loss functions for the discriminators (Equation 2) and the dense reward formulation (Equation 4). The theoretical justification for the bi-phasic training scheme is also well-articulated, explaining how it helps to overcome the exploration challenges in sparse reward settings.

---

> ### Author Rebuttal · Authors · 2025-03-31
>
> We thank the reviewer for their valuable feedback. We address your comments in the following.
>
> ---
>
> **Q: Discussion on limitations and future work**
> **A:** Thank you for pointing this out. In response, we have prepared an expanded discussion on limitations and future work. We will incorporate this expanded discussion into the final manuscript upon acceptance.
>
> *We acknowledge that our current implementation relies on demonstrations collected via RL-trained agents, which are typically well-aligned with task objectives but often unimodal and less representative of human-like variability. A key future direction is to extend DEMO3 to handle more **diverse and multimodal demonstrations**, such as those arising from motion planning, human teleoperation, or video sources. Robustness to such sources would broaden the applicability of our method to real-world data collection settings.*
>
> *Another limitation is that our current experiments are confined to **simulation environments**. We aim to deploy DEMO3 on **real robot hardware** to evaluate how well our dense reward learning and world model generalize under domain shift, sensor noise, and actuation delays, and whether the same constrained regime of demonstrations and interaction samples can yield reliable real-world policies. Notably, related work such as **MoDem** has already demonstrated that a similar pipeline can transfer successfully to real robots. Given that DEMO3 builds on and improves these components in simulation, we are optimistic that its benefits will carry over to physical systems.*
>
> *Finally, while our current **50% demonstration sampling ratio** already yields strong results, we believe that more **sophisticated sampling strategies** could further enhance learning efficiency. Inspired by prioritized replay buffers, we plan to explore **priority-based demonstration sampling** that adaptively focuses on more informative or rare transitions during training.*
>
> ---
>
> **Q: Thorough analysis of computational complexity**
> **A:** We appreciate the reviewer’s request for a clearer discussion of computational complexity. As shown in **Table 2** of the main paper, DEMO3 introduces **minimal computational overhead** compared to TD-MPC2. Specifically, our method adds only a lightweight MLP per stage (i.e., the **stage discriminators**), which are trained jointly with the world model. This design keeps the additional parameter count and forward/backward pass time inconsequential relative to the backbone model.
>
> Empirically, DEMO3 increases per-100k training time by only **6.7%** compared to TD-MPC2 (5.19h vs. 4.84h), and remains significantly faster than other demonstration-augmented baselines like **MoDem (8.37h)** and **LaNE (20.40h)**. These results suggest that DEMO3 offers strong performance gains with **minimal added training cost**, making it practical for real-world deployment scenarios where wall time is a limiting factor.
>
> We will make this comparison more explicit in the final version of the paper.
>
> ---
>
> **Q: Applying the method to a varying number of demonstrations**
> **A:** We thank the reviewer for raising this important point. We address this question in our **demonstration efficiency analysis** (Figure 8), where we evaluate performance as a function of demonstration count (5, 10, 25, 50, 100, 200) on two of the most challenging tasks: **StackCube** and **PegInsertion**. DEMO3 is the only method that consistently reaches the target success threshold with as few as **5 demonstrations**, and it continues to improve steadily with more data, **outperforming all baselines across the range**.
>
> For a more detailed view, we refer the reviewer to **Appendix A.2**, which includes **full learning curves under different demonstration regimes**. These results show that DEMO3 gracefully scales with more demonstrations while also being **highly effective in the low-data regime**.
>
> ---
>
> **Q: How sensitive is the approach to stage definition? How does it transfer to more “unstructured” tasks?**
> **A:** This is a great question. While DEMO3 is evaluated on tasks with explicit stage-wise structure, we believe that many real-world tasks — even those considered “unstructured” — can often be approximated with a **coarse stage decomposition**. For example, a task like human locomotion might involve an **initial acceleration phase** followed by a **steady gait phase**, which can naturally map to 2 distinct stages for learning purposes.
>
> To better understand the effect of stage granularity, we include a **reward granularity ablation (Figure 9)**. This experiment compares performance across **1, 2, 3, and fully dense stage definitions**. While performance improves with finer-grained supervision, **DEMO3 remains surprisingly robust even under coarse or minimal stage labeling**, performing competitively with human-engineered dense rewards using only 1-stage definitions.
>
> ---
>
> Please do not hesitate to let us know if you have any additional comments.

---

### Official Review · Reviewer_xwA3 · 2025-03-11

**Overall Recommendation:** 3

**Summary:**

This paper introduces a demonstration-augmented reinforcement learning method to solve data-efficient manipulation tasks with sparse rewards. By utilizing limited demonstrations, the policy, world model, and dense reward are effectively modeled, thus the long-horizon tasks can be solved in a multi-stage manner. Experiments on four benchmarks demonstrate the effectiveness of the proposed method.

## update after rebuttal
Thanks to the authors for your efforts during rebuttal. Most of my concerns are resolved. I have raised my rating.

**Claims And Evidence:**

Yes.

**Essential References Not Discussed:**

No.

**Experimental Designs Or Analyses:**

My major concerns lie in the experimental parts,

i) Only three existing works (i.e., TD-MPC2, Modem, LaNE) are used for comparisons,
  - the proposed method leverages the strength of Modem and TD-MPC2, thus the improvements over these two methods are a little bit trivial;
  - Why the previous SoTA (ie, LANE) performs well on Robosuite while performing extremely badly on the other three benchmarks (Fig. 1)?
  - since the method uses demonstrations for augmentation, existing imitation learning based methods should be compared;

ii) The tasks used for ablation are mainly chosen from ManiSkill, the environment where the method achieves the largest improvement over other methods. It is not convincing enough to evaluate the effectiveness of the proposed method;

**Methods And Evaluation Criteria:**

Not sufficiently. One main goal of this paper is to solve the sparse reward problem in long-horizon tasks. However, the reviewer finds the used benchmarks/datasets have a limited number of stages (i.e., <3).

**Other Comments Or Suggestions:**

No.

**Other Strengths And Weaknesses:**

No.

**Questions For Authors:**

Please see the experiment section.

**Relation To Broader Scientific Literature:**

Related.

**Theoretical Claims:**

No theoretical claims are made.

---

> ### Author Rebuttal · Authors · 2025-03-31
>
> We thank the reviewer for their valuable feedback. We address your comments below.
>
> ---
>
> **Q: Benchmarks/tasks have a limited number of stages**
> **A:** We appreciate the reviewer’s concern. While many tasks in our suite contain 2–3 defined stages, we argue that this already reflects significant temporal and compositional complexity, particularly in sparse reward visual manipulation. Prior works such as DrS, MoDem, and LaNE often evaluate on shorter or single-stage tasks. In contrast, DEMO3 handles longer-horizon behaviors with similarly sparse supervision, making our setup at least as challenging—if not more—than existing literature.
> That said, we agree that scaling DEMO3 to more complex, multi-stage tasks is a valuable next step. We are actively working on tasks like *Stack-N-Cubes*, requiring sequential completion of N > 3 subgoals. While we couldn’t finalize these in time for this submission, we plan to include preliminary results in the camera-ready version.
>
> ---
>
> **Q: Improvements over MoDem and TDMPC2 seem trivial since DEMO3 builds upon them**
> **A:** We thank the reviewer for this comment and appreciate the opportunity to clarify our benchmarking choices.. While DEMO3 does incorporate key ideas from both MoDem and TD-MPC2, we believe it is important—and indeed necessary—to benchmark directly against these methods to quantify the contribution of our core innovation: **dense, multi-stage reward learning**.
>
> TD-MPC2 and MoDem are **strong, state-of-the-art baselines** in visual model-based RL and demonstration-augmented RL, respectively. DEMO3 builds on their foundations but introduces a new training paradigm in which **reward, policy, and world model are trained jointly via online, stage-wise discriminator signals**, rather than using precomputed reward functions or task-specific shaping. This design results in a reward signal that evolves alongside the agent’s experience, improving both learning stability and sample efficiency.
>
> Rather than relying on a single benchmark, our results show that introducing learned reward components consistently improves performance across tasks and domains, as illustrated in Figures 5 and 7. Furthermore, our ablation studies demonstrate that the learned reward is particularly important in high-variance, long-horizon tasks where sparse rewards are insufficient for policy discovery.
>
> ---
>
> **Q: Comparing with Imitation Learning**
> **A:** Thank you for the suggestion. To complement comparisons with demonstration-augmented RL methods, we now include *Behavioral Cloning (BC)* results across all tasks. These are visible on the [project website](https://sites.google.com/view/icml2025demo3) and will be added to the final manuscript.
> As expected, BC performs better in simpler settings like Robosuite, but fails to succeed in harder domains like ManiSkill Manipulation, which involve longer horizons, multi-stage coordination, and high variability (see Appendix D.3). This supports the view that stronger supervision—via dense rewards or interaction—is needed in these benchmarks.
>
> ---
>
> **Q: Why does LaNE perform badly in ManiSkill and Meta-World?**
> **A:** LaNE was originally evaluated on Robosuite and may be tuned to that benchmark. We attempted to adapt its hyperparameters to all environments, but performance still varied widely. DEMO3, by contrast, uses the same hyperparameters across all benchmarks, demonstrating stronger out-of-the-box generalization.
> As shown in Appendix D.3, Robosuite is comparatively easier, while ManiSkill and Meta-World involve longer horizons, greater variability, and multimodal interactions—conditions under which LaNE’s nearest-neighbor mechanism struggles to scale.
> We also observed that LaNE performs well in early stages but often fails to progress further, aligning with its reliance on latent-space similarity. Reward plots (see LaNE Task Progress in [Additional Ablations](https://sites.google.com/view/icml2025demo3/additonal-ablations)) illustrate this behavior. We will clarify these findings in the final manuscript.
>
> ---
>
> **Q: ManiSkill ablations are not convincing enough**
> **A:** While main ablations focus on ManiSkill, *per-task results across all domains* are provided in Appendix A.1. We chose ManiSkill because it is the most challenging domain in our suite, requiring precise control, long-horizon planning, and generalization under randomization.
> BC performance also reflects this difficulty, performing modestly in Robosuite but struggling in ManiSkill. These characteristics make ManiSkill a valuable testbed for evaluating the impact of each component in DEMO3.
>
> That said, we agree that broader ablation coverage is useful. Additional ablations on Meta-World are available here: [Meta-World Ablations](https://sites.google.com/view/icml2025demo3/additonal-ablations). These show trends consistent with those in ManiSkill and support the generality of our framework.
>
> ---
>
> Please don’t hesitate to reach out with further comments.

---

### Official Review · Reviewer_wCad · 2025-03-12

**Overall Recommendation:** 4

**Summary:**

This paper proposes DEMO, a framework that learns dense rewards from demonstrations and interactions with the environment to aid model-based RL learning. DEMO uses a multi-stage paradigm where it learns dense rewards from sparse reward signals (from stage indicators) to indicate "progress" and uses the learned dense reward to better learn policies. DEMO learns the reward model from visual signals and shows that incorporating the DEMO reward can help policies learn faster and converge to better performances. Experiments on a number of popular benchmarks (ManiSkill, Meta-World, Robotuite) show that the proposed method achieves SOTA performance using fewer samples.

**Claims And Evidence:**

This work claims that using a small number of samples and learned dense rewards can speed up and help learning long-horizon tasks using MBRL. The results on the benchmarks and ablations verify these claims. Especially on the harder tasks (in ablation A.2), DEMO achieves the best results with a small number of demonstrations.

**Essential References Not Discussed:**

References are adequate.

**Experimental Designs Or Analyses:**

This work showcases results on long-term tasks such as "stack cube" and "stick pull" that best demonstrate the benefit of having multi-stage rewards and learned dense rewards. DEMO achieves the best results. These tasks can best demonstrate the benefit of using learned dense rewards as they are challenging and each step leads to another.

In terms of analysis, I feel like adequate analysis is provided in terms of performance of the overall results. Ablation also shows that each component is important for achieving the best result. Some missed opportunities in verifying that the learned dense reward actually corresponds to "task progress" as claimed in the paper.

**Methods And Evaluation Criteria:**

Yes, the proposed benchmarks and evaluation criteria are suitable for the application and problem setup. ManiSkill and MetaWorlds are challenging and popular benchmarks in this space, and the proposed method achieves better performance than baselines.

No real-robot experiments are conducted, though, which would help bolster the claims of the method.

**Other Comments Or Suggestions:**

I would suggest including live plots of the learned reward function and videos of the policy rollout to better showcase the capabilities of the learned reward function and policy.

**Other Strengths And Weaknesses:**

## Additional Strength
- Overall, I think this paper provides an interesting and novel idea in dense reward learning from demonstrations and interactions with the environment.
- Analysis of the reward granularity shows that the learned reward signals are near optimal compared to dense rewards engineered by humans.

**Questions For Authors:**

None

**Relation To Broader Scientific Literature:**

I think this work fits nicely in the model-based RL framework and the behavior cloning literature where the strengths of both fields are combined. If a few demonstrations can be effectively used to learn dense rewards for multi-stage and long-horizon tasks, then the benefit of using RL and interactive labels can be maximally leveraged for future results.

**Theoretical Claims:**

No theoretical claims are proposed in this work.

---

> ### Author Rebuttal · Authors · 2025-03-31
>
> We thank the reviewer for their valuable feedback. We address your comments in the following.
>
> ---
>
> **Q: Suggestion about live plots with policy rollout (1 to 1 video with reward evolution and policy rollout)**
> **A:** We fully agree with the reviewer that it is important to verify whether the learned dense reward accurately reflects task progress, especially since this is central to our method. Following this suggestion, we have added live plots of the learned reward function aligned with video rollouts, now available in the **“Learned Dense Reward”** section of the project website: [https://sites.google.com/view/icml2025demo3](https://sites.google.com/view/icml2025demo3).
>
> To provide a comprehensive overview, we include three representative cases:
> - a **successful rollout**,
> - a **failed rollout**, and
> - a **semi-successful rollout** that illustrates fluctuating behavior, alternating between progress and regression in the task.
>
> These visualizations clearly demonstrate that the learned reward function captures nuanced task progression, aligning with intuitive notions of success, failure, and partial recovery. We believe this addition strongly supports the claim that our reward model tracks progress in long-horizon, multi-stage tasks.
>
> ---
>
> **Q: Real robot experiments**
> **A:** We acknowledge that validating DEMO3 in real hardware would further strengthen our contributions, and we plan to pursue this in future work. We are particularly interested in testing whether DEMO3 retains its strong sample efficiency and robustness under real-world conditions. Notably, prior work such as **MoDem-v2** has already demonstrated that a similar demonstration-augmented model-based pipeline can be successfully transferred to real robots. Given that DEMO3 builds on and improves these components, we are optimistic that its benefits will carry over to real-world deployment as well.
>
> ---
>
> Please do not hesitate to let us know if you have any additional comments.

---

> > ### Comment · Reviewer_wCad · 2025-04-06
> >
> > Thank you for the response. I appreciate and enjoy the live plots. They demonstrate the learned reward functions well. I will maintain my acceptance rating.

---

### Decision · Program_Chairs · 2025-05-01

**Decision:**

Accept (poster)

**Comment:**

The paper proposes demo-augmented reward, policy and world model learning. This framework exploits the coherent learning among these elements for efficient learning from visual inputs. The paper is well motivated, clear written and with comprehensive experiments. All three reviewers reach concensus that the paper is ready for publication.

Please incorporate all the necessary revisions into the camera ready version, including:
- technical details / clarification
- limitation on failure cases / real-robot experiments
- ablations as per requested by reviewers in the rebuttal.